# Structure-aware Granular-Ball based Information Bottleneck for Multi-modal Clustering

**Zhengzheng Lou** [1]  **Yuhan Zhan** [1]  **Mingyang Lv** [1]  **Yingxuan Li** [1]  **Yuyang Du** [1]  **Shizhe Hu** [1]

## Abstract

Multi-modal clustering, which integrates information from diverse sources and feature modalities, has shown great potential in data mining and computer vision. However, existing methods relying on single-granularity relationships often struggle with complex data distributions, leading to limited performance, as fine-grained features are prone to local heterogeneity and redundant perturbations while coarse-grained representations tend to lose local structural information. To address these limitations, we introduce granular-balls (GBs), adaptive multi-granularity hyperspheres that enclose similar samples, and propose the Structure-aware Granular-Ball based Information Bottleneck (SGB-IB) algorithm. This method initializes the dataset as a single GB and recursively splits GBs based on a purity metric, which quantifies the average mutual information between sample features and K-means-derived pseudo-labels across all modalities. It also balances local structure preservation and global redundancy suppression through a structure-aware objective function. Extensive experiments on benchmark datasets demonstrate that our method outperforms state-of-the-art approaches, validating the effectiveness of fusing GB structures with information-theoretic principles.

## 1. Introduction

In the real world, research subjects are often characterized by multi-modal information, where the same entity is described by data from diverse modalities (e.g., images, text, video) (Hao et al., 2025; Bica et al., 2024; Pan et al., 2025). Multi-modal clustering (MMC) aims to discover cross-modal consensus while retaining discriminative features, and has been widely applied in diverse real-world scenarios, including medical analysis, visual recognition, and multi-modal data mining (Dornaika et al., 2024; Mao et al., 2021; Liu et al., 2025).

Existing MMC methods fall into three main paradigms: (1) Graph-based methods integrate cross-modal information via graph learning, including vector quantization with self-regularization to reduce complexity (Cai et al., 2024), and modality-aware edge reweighting for adaptive graph structure alignment (He et al., 2025); (2) Matrix decomposition-based methods learn shared latent factors through non-negative or low-rank decomposition, such as multi-view non-negative matrix factorization for complementary information fusion (Khalafaoui et al., 2022), and subspace clustering for hyperspectral-lidar fusion (Yu et al., 2025); (3) Feature-based methods mine cross-modal associations via subspace mapping or information metrics. Typical studies include multi-view kernel modeling (Liu et al., 2023), contrastive learning based embedding optimization (Chen et al., 2021), and differentiable information bottleneck (IB) models for unified feature compression (Yan et al., 2024).

However, two critical limitations plague MMC methods, particularly for heterogeneous multi-modal data. First, taking individual samples as the basic input unit overemphasizes single-instance characteristics, amplifying modality-specific local variations and hindering the modeling of global cross-modal structural consistency. Second, fixed single-granularity representations lack flexibility to adjust weights according to local structural quality, failing to balance global cross-modal consensus and fine-grained intra-modal discriminative ability.

To address these issues, we draw inspiration from the granular-ball (GB) paradigm (Xia et al., 2025), which groups similar samples into homogeneous clusters via a macro-first cognitive model. By shifting from sample-level to GB-level representations, GB inherently preserves global structural consistency and suppresses modality-specific perturbations. We propose a novel MMC algorithm called Structure-aware Granular-Ball based Information Bottleneck (SGB-IB), which redefines MMC by taking GBs as fundamental processing units in the IB framework (Lou

[1]School of Computer and Artificial Intelligence, Zhengzhou University, Zhengzhou, China. Correspondence to: Shizhe Hu < ieshizhehu@gmail.com, https://shizhehu.github.io/>.

*Proceedings of the 43$^{rd}$ International Conference on Machine Learning*, Seoul, South Korea. PMLR 306, 2026. Copyright 2026 by the author(s).

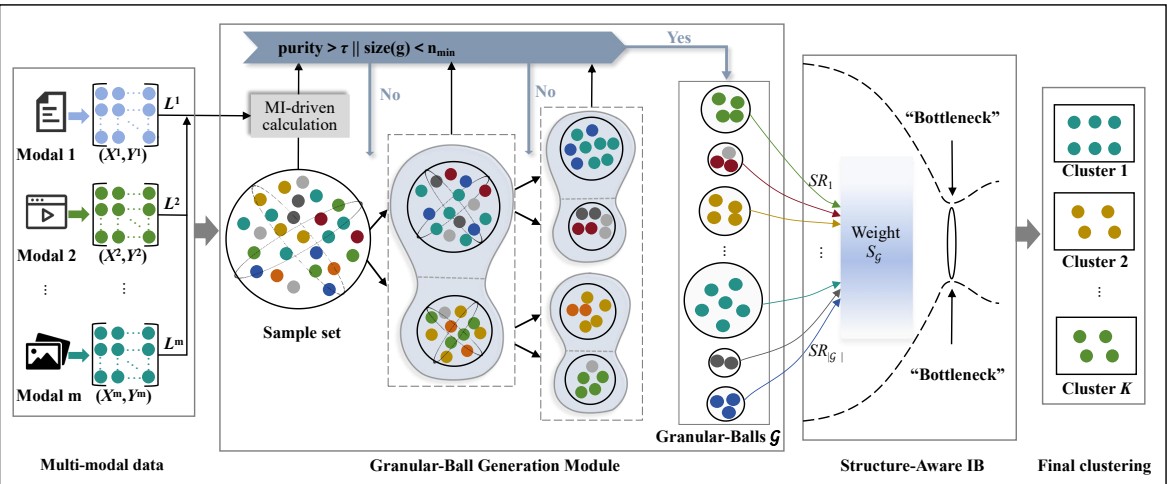

*Figure 1.* The framework of the SGB-IB method consists of four sequential modules. First, the Multi-Modal Input module processes diverse multi-modal features and generates pseudo-labels for samples via K-means, which are used to compute GB purity in subsequent steps. The module then constructs the initial multi-modal sample set. Second, Granular-Ball Construction recursively splits the multi-modal sample set into GB units in a binary manner; the splitting terminates when a GB satisfies the purity threshold $\tau$ or its sample size falls below the minimum requirement $n_{\min}$. Third, Structure-aware IB leverages the constructed GB units to impose dynamic weight constraints on the IB objective function based on the GB similarity ratio, balancing the preservation of cross-modal discriminative information and the suppression of intra-modal redundant information. Finally, Clustering Optimization produces the clustering results by making optimal cluster assignment decisions for GB units based on the learned representations.

et al., 2013) and designing dynamic information preservation weights adapted to GB structural similarity. Specifically, SGB-IB introduces two key innovations: GB features are used for both compression and preservation, where intra-GB consistency filters local perturbations and retains global topology via group-wise representation; dynamic information preservation weights are computed using the GB-target cluster similarity ratio (SR), explicitly capturing coarse-grained cross-modal consensus and fine-grained intra-modal details. Experimental results on public datasets demonstrate that SGB-IB outperforms the latest MMC advances. Our main contributions are summarized as follows:

- We propose a new MMC framework by unifying GB representations with the IB principle, which extends sample-level clustering to the GB level for stable and scalable multi-modal information compression.

- We develop an information-driven GB generation scheme with constrained recursive binary splitting to enable data-adaptive granularity refinement for diverse multi-modal data distributions.

- We design a structure-aware weighting mechanism and a GB-constrained IB objective, which optimize the compression-preservation by leveraging GB structural similarity, while improving cross-modal semantic alignment for discriminative representation learning.

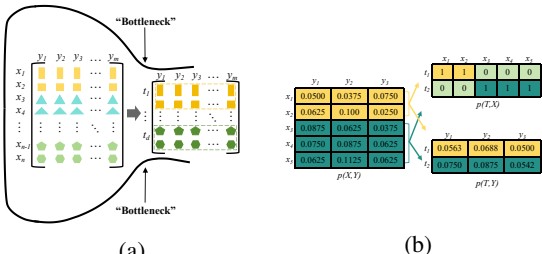

(a)                                    (b)

*Figure 2.* Information Bottleneck model. (a) The process of learning a compact representation $T$ from input $X$ while maximizing information about target $Y$; (b) The joint probability distributions underlying the IB optimization, including $p(X, Y)$, $p(T, Y)$, and $p(T, X)$.

## 2. Prior Knowledge

IB (Tishby et al., 1999) is an information-theoretic framework widely used in real-world tasks, such as unsupervised image segmentation (Bardera et al., 2009) and pattern classification (Yan et al., 2020), with a systematic overview of its theoretical extensions and applications in (Hu et al., 2024). As illustrated in Figure 2(a), the core goal of IB is to learn a compact bottleneck representation $T$ for input $X$ that maximizes information retained about $Y$. This process relies on the joint probability distributions shown in Figure 2(b), including $p(X, Y)$, $p(T, Y)$, and $p(T, X)$. This objective is formalized as:

$$\mathcal{L}_{IB} = I(T; Y) - \beta^{-1} I(T; X). \tag{1}$$

| | |
|---|---|
| $m$ | Number of modalities |
| $n$ | Number of samples |
| $K$ | Number of clusters |
| $T$ | Clustering structure |
| $X^i$ | Samples of the $i$-th modality |
| $Y^i$ | Feature representation of the $i$-th modality |
| $L^i$ | Pseudo-labels of the $i$-th modality |
| $c_g^i$ | Center of GB $g$ in the $i$-th modality |
| purity | Multi-modal purity of a GB |
| SR | Structural similarity ratio of a GB |
| $S_{\mathcal{G}}$ | Structure-aware weight for GB set $\mathcal{G}$ |
| $\alpha, \beta$ | Trade-off hyperparameters |
| $\tau, n_{\min}$ | Purity and size thresholds for recursive GB splitting |
| $|\mathcal{G}|$ | Total number of GBs |

Here, $I(T; X)$ quantifies compression, $I(T; Y)$ measures relevance between $T$ and $Y$, and $\beta > 0$ is a Lagrange multiplier regulating the trade-off between compression and information preservation. This core mechanism of IB motivates our structure-aware extension, which integrates GB into IB to address the fixed-granularity limitation in MMC methods. Existing IB methods operate solely on individual samples, facing two key issues: an inability to balance global consistency and fine-grained discriminability, and an over-reliance on single-sample quality due to a lack of coarse-level structuring.

GB is a coarse-grained data unit that partitions similar samples into homogeneous groups, serving as a basic structural element for multi-modal modeling. Unlike capacity expansion, such granularization acts as a constrained simplification that improves generalization by aggregating local samples into stable units, thereby reducing variance from individual perturbations (Xia et al., 2019).

To mitigate these limitations, we equip IB with GB-based structural constraints and adaptive weighting mechanisms tailored to MMC challenges.

# 3. The Proposed Method

## 3.1. Problem Formulation

Given a multi-modal dataset $\mathcal{X}$, denoted by $\{X^i\}_{i=1}^m$, there are $m$ modalities leading to variables $\{Y^i\}_{i=1}^m$, where $Y^i \in \mathbb{R}^{n \times d^i}$ denotes the feature matrix of the $i$-th modality (where $n$ is the total number of samples and $d^i$ is the feature dimension). The goal of MMC is to partition $\mathcal{X}$ into $K$ clusters $T = \{t_1, t_2, \ldots, t_K\}$. To achieve this, we in-

troduce a GB representation $\mathcal{G} = \{g_1, g_2, \ldots, g_{|\mathcal{G}|}\}$, where each GB $g$ comprises a subset of samples and is characterized by its multi-modal center $\mathcal{C}_g = \{c_g^1, \ldots, c_g^m\}$. This GB representation acts as the fundamental unit for subsequent computation and optimization, bridging individual samples and the final clustering result. To enhance clarity and conciseness, we summarize the main notations in Table 1.

## 3.2. Granular-Ball Generation

This module aims to construct a set of GBs $\mathcal{G}$ that preserves local data structures while reducing sample-level redundancy. We propose an information-theoretic purity measure to drive a recursive splitting process, ensuring each GB maintains internal consistency both within and across modalities. This focus on intrinsic structure provides a solid foundation for subsequent MMC tasks.

Initially, the entire dataset is treated as a single GB. For each modality $i$, we generate $K$-class pseudo-labels via K-means clustering, denoted as $L^i$. Following existing GB research (Cheng et al., 2026; 2024), these pseudo-labels serve solely for coarse-grained initialization; the subsequent dual-threshold partitioning and IB optimization jointly mitigate the impact of initial label deviations. The multi-modal purity of a GB $g$ is then defined as the average mutual information (MI) between sample features and their corresponding pseudo-labels within $g$ across all modalities:

$$\text{purity}_g = \frac{1}{m} \sum_{i=1}^m \frac{1}{|g|} \sum_{x \in g} I(y_x^i; l_x^i), \qquad (2)$$

where $y_x^i$ and $l_x^i$ are the feature and pseudo-label of sample $x$ in $g$ for modality $i$, and $|g|$ denotes the number of samples in GB $g$. This metric integrates MI across modalities to characterize intra-GB compactness and consistency. With this purity metric, we can recursively partition the dataset into GBs of adaptive granularities.

Each GB is recursively split via K-means into two subsets if its purity is below a predefined threshold $\tau$ and its size exceeds a minimum sample size $n_{\min}$, otherwise it is retained in the final GB set $\mathcal{G}$. This splitting process iterates until all GBs satisfy the termination criteria, yielding a data-adaptive multi-granularity representation.

The dual-threshold scheme serves distinct yet coordinated purposes: the purity threshold promotes the internal structural consistency of each GB, while the minimum sample size threshold limits partitioning granularity to avoid over-splitting into fragmented GBs with sparse or even single samples. Together, they achieve a reasonable balance between structural quality and partition granularity.

For each GB $g \in \mathcal{G}$, the $i$-th modal center is defined as the

mean of all samples in $g$, as follows:

$$c_g^i = \frac{1}{|g|} \sum_{x \in g} x^i. \tag{3}$$

Integrating all modal centers yields $\mathcal{C}_g = \{c_g^1, c_g^2, \ldots, c_g^m\}$. This representation encapsulates cross-modal consensus and provides a stable foundation for computing structure-aware weights and information terms, thereby forming a closed technical loop between GB generation and subsequent optimization.

### 3.3. Overview of Objective Function

Building on GB generation, we enhance the IB framework with three key components: structure-aware weighting via $\mathrm{SR}_g$, multi-granularity representation from GBs, and cross-modal consensus learning by structural constraints. This augmentation allows us to optimize the trade-off between information preservation, structural consistency, and data compression at the GB level. The objective function is formulated as:

$$\mathcal{L} = S_{\mathcal{G}} \cdot \sum_{i=1}^{m} I(T; Y^i) - \beta^{-1} \sum_{i=1}^{m} I(T; X^i), \tag{4}$$

where $\sum_{i=1}^{m} I(T; Y^i)$ denotes the information preservation term, retaining cross-modal discriminative information from features; $S_{\mathcal{G}}$ is the structure-aware weight that optimizes modal weights to strengthen inter-modal consensus; and $\sum_{i=1}^{m} I(T; X^i)$ represents the information compression term, which reduces raw data redundancy via intra-GB redundant feature filtering. $\beta > 0$ is a trade-off hyperparameter regulating compression and preservation.

This objective integrates the structure-aware weight $S_{\mathcal{G}}$ into the IB optimization to enable adaptive trade-off regulation. With GBs as the information carrier, these three components jointly constitute a complete optimization paradigm: the preservation module extracts discriminative information from GB features; the constraint module reinforces inter-modal consensus via $S_{\mathcal{G}}$; the compression module filters redundancy, ultimately achieving superior performance in MMC.

### 3.4. Information Preservation

The information preservation term aims to maximize the relevant information from each modality in the clustering structure $T$. Unlike traditional methods that directly use individual sample features for this purpose, our approach optimizes the process via the GB framework. It offers two core advantages: First, GBs aggregate local samples into stable units, reducing outlier interference and thereby enhancing the reliability of information preservation. Second, leveraging their inherent multi-granularity structure

(i.e., GBs of varying sizes), each GB's multi-modal center naturally serves as a unified statistical representation for heterogeneous raw features. The consistent mean-based calculation across all modalities ensures stable and comparable computation of $\sum_{i=1}^{m} I(T; Y^i)$.

By maximizing $\sum_{i=1}^{m} I(T; Y^i)$, our method strengthens the correlation between the clustering structure $T$ and the relevant information of each modality $Y^i$. This allows $T$ to accurately capture both inter-modal commonalities and intra-modal specificities.

### 3.5. Structure-aware Granular-Ball Constraints

Not all GBs contribute equally to global clustering. High-quality GBs with strong cross-modal consistency drive reliable clustering, while low-quality ones (statistically inconsistent or misassigned) reduce stability. To address this, we introduce a structure-aware weight $S_{\mathcal{G}}$ into the IB objective. The core intuition is to assign differentiated weighting contributions to GBs based on their structural alignment with clusters, thereby prioritizing high-quality structural information and enhancing clustering stability.

The core idea of structure awareness is to quantify the alignment between GBs' local structures and the global clustering via SR. Following common practice, we adopt cosine similarity to measure pairwise GB similarity. For a GB $g \in \mathcal{G}$, we define the average multi-modal similarity $\mathrm{sim}(g, t)$ between $g$ and its assigned cluster $t$. The multi-modal center of $g$ is denoted by $\mathcal{C}_g$, and its $i$-th modal component is $c_g^i$. $\mathrm{sim}(g, t)$ computes the average of cosine similarities between $\mathcal{C}_g$ and the multi-modal centers of all GBs in $t$:

$$\mathrm{sim}(g, t) = \frac{1}{|t|} \sum_{g' \in t} \left( \frac{1}{m} \sum_{i=1}^{m} s\left(c_g^i, c_{g'}^i\right) \right), \tag{5}$$

where $s(\cdot, \cdot)$ denotes the cosine similarity between two modal centers. Based on $\mathrm{sim}(g, t)$, we define $\mathrm{SR}_g$ to quantify $g$'s structural alignment with the global clustering, which serves as the fundamental constituent unit of the global weighting factor $S_{\mathcal{G}}$. Mathematically, $\mathrm{SR}_g$ is formulated as:

$$\mathrm{SR}_g = \frac{\mathrm{sim}(g, t)}{\frac{1}{K-1} \sum_{t' \neq t} \mathrm{sim}(g, t')}, \tag{6}$$

where the denominator $\frac{1}{K-1} \sum_{t' \neq t} \mathrm{sim}(g, t')$ denotes the average similarity between $g$ and GBs in the other $K - 1$ clusters. A value of $\mathrm{SR}_g > 1$ indicates that $g$ is a high-quality GB (it is more aligned with $t$ than with any other cluster); conversely, $\mathrm{SR}_g \leq 1$ labels $g$ as a low-quality GB with weak structural consistency. Based on $\mathrm{SR}_g$, we design the structure-aware weight as:

$$S_{\mathcal{G}} = \sum_{g=1}^{|\mathcal{G}|} \left(1 - \alpha \left(1 - e^{-\mathrm{SR}_g}\right)\right), \tag{7}$$

where $\alpha \in [0, 1]$ controls the strength of the structural quality constraint. When $\alpha = 0$, $S_{\mathcal{G}} = |\mathcal{G}|$ with uniform contribution from all GBs, disabling the quality-aware prioritization mechanism for $\sum_{i=1}^{m} I(T; Y^i)$. When $\alpha = 1$, Eq. (7) simplifies to $S_{\mathcal{G}} = \sum_{g=1}^{|\mathcal{G}|} e^{-\text{SR}_g}$.

This design establishes an inverse weighting mechanism: high-quality GBs (large $\text{SR}_g$) yield small contributions to $S_{\mathcal{G}}$ via the exponential decay term $e^{-\text{SR}_g}$, while their stronger cross-modal consistency enables them to capture more discriminative information in $\sum_{i=1}^{m} I(T; Y^i)$. Conversely, low-quality GBs contribute more to $S_{\mathcal{G}}$ but carry limited discriminative information. This inherent asymmetry yields a complementary effect within the product $S_{\mathcal{G}} \cdot \sum_{i=1}^{m} I(T; Y^i)$, guiding optimization toward regions where structural coherence and discriminative information jointly concentrate, thereby improving clustering performance.

### 3.6. Information Compression

The information compression term aims to maximize the compression of redundant information while preserving discriminative information, where $\sum_{i=1}^{m} X^i$ denotes input GB representations. A key innovation lies in integrating GB structural properties into the IB compression process: we enhance the compression granularity from the sample level to the GB level, thereby utilizing GBs' local coherence to mitigate sample-specific inconsistencies and strengthen the filtering of irrelevant information specific to each modality. This design is theoretically justified by the intrinsic local coherence of GBs, which inherently supports the natural clustering tendencies of the data and reduces the impact of fine-grained variations in individual samples. To formalize this GB-level compression, the MI is defined as:

$$\sum_{i=1}^{m} I(T; X^i) = \sum_{g \in \mathcal{G}} I(T; \mathcal{C}_g), \quad (8)$$

where samples are processed into GBs, and $I(T; \mathcal{C}_g)$ quantifies the association between $T$ and GB $g$.

The hyperparameter $\beta$ balances information preservation and compression. When $\beta$ approaches 0, compression is prioritized, retaining only the essential cross-modal structural information. As $\beta$ increases, the compression constraint is relaxed, enabling the clustering model to capture finer GB-level discriminative details that are critical for separating subtle inter-modal patterns.

### 3.7. Optimization

To find an optimal solution for the proposed SGB-IB, we iteratively adjust GB-cluster affiliations via a sequential extract-merge process. Details are provided in Algorithm 1.

**Algorithm 1** Algorithm for Optimizing the SGB-IB

1: **Input**: Multi-modal data $\{X^i\}_{i=1}^{m}$, number of clusters $K$, hyperparameters $\alpha, \beta$, thresholds $n_{\min}$ and $\tau$.
2: **Output**: Final clustering results.
3: Initialize all samples as one single GB.
4: Perform recursive binary splitting until the final GB set $\mathcal{G} = \{g_1, g_2, \ldots, g_{|\mathcal{G}|}\}$ is obtained.
5: Randomly partition the GB set $\mathcal{G}$ into $K$ clusters.
6: **repeat**
7:     Recompute $S_{\mathcal{G}}$ via Eq. (7).
8:     **for** each GB $g$ **do**
9:         Extract $g$ from $t^{\text{old}}$ to form a singleton cluster $\{g\}$.
10:         Compute the optimal target cluster $t^{\text{new}} = \arg\min_{t \in T} \Delta\mathcal{L}(\{g\}, t)$.
11:         Merge $g$ into $t^{\text{new}}$ (or back into $t^{\text{old}}$ if unchanged).
12:     **end for**
13: **until** Data partition unchanged or a fixed number of iterations finished.
14: **Return** the final clustering results.

The GB set $\mathcal{G}$ is first randomly partitioned into $K$ clusters. For each $g \in \mathcal{G}$, $g$ is extracted from its original cluster $t^{\text{old}}$ to form a singleton cluster $\{g\}$ and temporarily increasing the cluster count to $K + 1$. It is then merged into the target cluster $t^{\text{new}} \in T$ to maintain the $K$-cluster constraint.

Each extract-merge step minimizes the merger cost $\Delta\mathcal{L}(\{g\}, t)$ to maximize the objective $\mathcal{L}$, where $t^{\text{new}} = \arg\min_{t \in T} \Delta\mathcal{L}(\{g\}, t)$, thus ensuring monotonic growth of $\mathcal{L}$ and avoiding ineffective adjustments. The merger cost quantifies the objective change before/after merging $g$ into $t$, with $S_{\mathcal{G}}$ (a global structure-aware factor) fixed per merger for stability and re-computed via Eq. (7) after each full extract-merge iteration over all GBs. The cost formula is defined as follows:

$$\Delta\mathcal{L} = S_{\mathcal{G}} \cdot \sum_{i=1}^{m} \Delta I(T; Y^i) - \beta^{-1} \sum_{i=1}^{m} \Delta I(T; X^i). \quad (9)$$

We estimate the compression term $\Delta \sum_{i=1}^{m} I(T; X^i)$ via MI over GBs. Using the chain rule of MI, this term can be expanded to:

$$\sum_{i=1}^{m} I(T; X^i) = \sum_{i=1}^{m} \sum_{t=1}^{K} p(t) \sum_{g \in t} D_{\text{KL}}\left(p(c_g^i \mid t) \parallel p(c_g^i)\right) \quad (10)$$

where $D_{\text{KL}}(\cdot \parallel \cdot)$ denotes the Kullback-Leibler (KL) divergence (Cover & Thomas, 2005). Let $T^{\text{old}}$ and $T^{\text{new}}$ denote the clustering structures before and after merging. For modality $i$, merging $g$ into $t$ is defined as:

$$\begin{cases} p(\tilde{t}) = p(c_g^i) + p(t) \\ p(y^i \mid \tilde{t}) = \pi_1 \cdot p(y^i \mid c_g^i) + \pi_2 \cdot p(y^i \mid t), \end{cases} \quad (11)$$

where $\pi_1$ and $\pi_2$ are probability weights that satisfy:

$$\pi = \{\pi_1, \pi_2\} = \left\{ \frac{p(c_g^i)}{p(\tilde{t})}, \frac{p(t)}{p(\tilde{t})} \right\}. \quad (12)$$

For the $i$-th modality, the change in the information preservation term before and after merging can be expanded using the Jensen-Shannon (JS) divergence, which is defined as:

$$\begin{aligned}
\Delta I(T; Y^i) &= I(T^{\text{old}}; Y^i) - I(T^{\text{new}}; Y^i) \\
&= p(c_g^i) \sum_{y^i} p(y^i \mid c_g^i) \log \frac{p(y^i \mid c_g^i)}{p(y^i \mid \tilde{t})} \\
&\quad + p(t) \sum_{y^i} p(y^i \mid t) \log \frac{p(y^i \mid t)}{p(y^i \mid \tilde{t})}.
\end{aligned} \quad (13)$$

Substituting Eq. (11) and simplifying, we obtain:

$$\Delta I(T; Y^i) = p(\tilde{t}) \cdot JS\big[p(y^i \mid c_g^i), p(y^i \mid t)\big], \quad (14)$$

Similarly, we derive $\Delta I(T; X^i)$ as follows:

$$\Delta I(T; X^i) = p(\tilde{t}) \cdot JS\big[p(c_g^i), p(c_g^i \mid \tilde{t})\big]. \quad (15)$$

Combining this with the structure-aware weight $S_{\mathcal{G}}$ yields the final merger cost as follows:

$$\begin{aligned}
\Delta \mathcal{L}^i(\{g\}, t) = p(\tilde{t}) \cdot \big( &S_{\mathcal{G}} \cdot JS\big[p(y^i \mid c_g^i), p(y^i \mid t)\big] \\
&- \beta^{-1} \cdot JS\big[p(c_g^i), p(c_g^i \mid \tilde{t})\big]\big).
\end{aligned} \quad (16)$$

### 3.8. Differences with Related Methods

Existing IB methods typically operate at the fine-grained sample level, modeling and compressing features for individual samples. Consequently, this makes them susceptible to local distribution variations and modality-specific perturbations. In contrast, SGB-IB elevates the basic modeling unit of the IB framework from individual samples to GBs. This allows our method to inherit the GB's distinct strength in representing coarse-grained structures while leveraging the IB principle to perform discriminative information preservation and redundancy compression at this GB-level, effectively unifying GB learning with information-theoretic optimization.

Standard GB-based clustering methods optimize clustering using geometric criteria such as purity and distance, without explicit information-theoretic grounding to quantify representation informativeness. SGB-IB addresses this limitation by embedding structural quality into the information-theoretic objective via structure-aware weights. This achieves a deep coupling between structural awareness and information-theoretic optimization, thereby enabling adaptive coarse-grained representation learning within the IB framework.

Most existing hierarchical and structured GB approaches adopt graph modeling or fixed granularity partitioning schemes (Cheng et al., 2026; Jia et al., 2025). Such designs rely on heuristics or empirical rules and lack an information-theoretic mechanism for adaptive granularity control. In contrast, SGB-IB unifies GB structure awareness and the IB principle through information-driven recursive splitting. The splitting process employs MI-driven purity as the core criterion, enabling granularity to adapt dynamically to the data's intrinsic information structure.

### 3.9. Complexity Analysis

Our algorithm consists of two stages: GB generation via recursive binary splitting and iterative optimization. For $n$ samples with total dimensionality $D = \sum_{i=1}^m |Y^i|$ across $m$ modalities, GB generation costs $O(nD \log |\mathcal{G}|)$. Each optimization iteration computes structure-aware weights via pairwise GB similarities, with a time complexity of $O(|\mathcal{G}|^2 D)$ $O(|\mathcal{G}|^2 D)$, and merger costs at $O(|\mathcal{G}|KD)$ (dominated by the former when $|\mathcal{G}| \gg K$). The total time complexity over $L$ iterations until convergence is $O(nD \log |\mathcal{G}| + L|\mathcal{G}|^2 D)$.

## 4. Experiment

### 4.1. Datasets

We evaluate our method on five widely used MMC datasets, described as follows. **SOCCER** comprises 280 web-collected soccer team images, with 40 samples per class across 7 categories. Inter-class visual similarity poses notable clustering challenges, and the dataset is represented by three visual modalities. **BBC** (Greene & Cunningham, 2006) is collected from the official BBC website and comprises 4 modalities, each containing 685 news articles. **Reuters** (Amini et al., 2009) contains 1,200 multilingual documents with pairwise translations evenly distributed across 6 categories. Each document is available in English, French, and German, forming three independent modalities that capture cross-lingual semantic information. **WVU** (Laptev, 2005) comprises 10 classes of human action videos captured from diverse perspectives. For experimental purposes, non-adjacent modalities (1, 3, 5, 7) are employed, simulating real-world scenarios such as road surveillance. **COIL100** is a large-scale visual dataset with 7,200 images of 100 objects. Images of each object are captured via a motorized turntable. Two visual modalities, SIFT (Lowe, 2004) and SURF (Nilsback & Zisserman, 2008), are used for multi-modal representation.

### 4.2. Compared Methods

We compare SGB-IB against 15 representative state-of-the-art methods across three categories of clustering approaches

*Table 2.* Clustering performance on various datasets (bold denotes the best performance, underline denotes the second best).

| Methods | SOCCER | | Reuters | | BBC | | WVU | | COIL100 | |
|---|---|---|---|---|---|---|---|---|---|---|
| | ACC | NMI | ACC | NMI | ACC | NMI | ACC | NMI | ACC | NMI |
| KM | 26.24±5.0 | 19.37±7.2 | 25.22±6.8 | 10.89±7.1 | 37.80±4.8 | 11.59±8.2 | 27.42±4.3 | 33.87±5.9 | 27.58±1.7 | 57.57±2.0 |
| IB | 46.74±1.3 | 44.38±2.1 | 41.98±3.2 | 25.22±2.4 | 72.90±9.0 | 60.32±6.3 | 52.23±2.1 | 50.75±0.7 | 40.85±1.6 | 68.54±0.8 |
| AmKM | 21.73±2.1 | 7.06±2.3 | 21.50±6.0 | 4.73±5.3 | 40.73±6.7 | 12.79±11.0 | 27.22±5.4 | 24.28±7.0 | 30.23±3.4 | 51.56±3.0 |
| AmIB | 50.72±1.6 | 45.12±1.3 | 29.25±2.4 | 19.93±1.9 | 79.27±8.2 | 72.75±7.3 | 53.48±1.6 | 51.37±0.4 | 38.89±2.1 | 67.94±2.5 |
| LMSC | 47.31±6.3 | 44.56±5.1 | 47.39±3.1 | 27.48±2.3 | 71.96±6.4 | 53.21±5.3 | 57.31±4.2 | 58.68±3.0 | 50.37±0.8 | 67.97±0.3 |
| MLAN | 28.76±2.2 | 21.23±3.6 | 21.32±0.0 | 8.13±0.0 | 69.23±7.4 | 49.02±5.6 | 37.82±0.0 | 34.96±0.0 | 44.35±1.1 | 61.36±1.0 |
| GMC | 29.29±0.0 | 25.82±0.0 | 19.92±0.0 | 13.66±0.0 | 69.34±0.0 | 56.28±0.0 | 46.78±0.0 | 54.92±0.0 | 37.60±0.0 | 47.03±0.0 |
| SMVSC | 28.03±0.0 | 9.29±0.0 | 49.21±2.2 | 30.32±2.0 | 53.52±4.3 | 29.98±3.8 | 45.43±2.7 | 44.89±2.2 | 50.33±1.2 | **70.13±0.8** |
| FPMVS-CAG | 22.14±0.0 | 5.29±0.0 | 46.50±0.0 | 24.69±0.0 | 79.50±0.0 | 72.15±0.0 | 49.21±0.0 | 49.52±0.0 | 48.72±0.0 | 69.61±0.0 |
| OMSC | 24.29±0.0 | 6.70±0.0 | 50.58±0.0 | 25.62±0.0 | 65.11±0.0 | 54.86±0.0 | 52.69±0.0 | 56.22±0.0 | 48.10±0.0 | 67.77±0.0 |
| FIMVC-VIA | 50.81±0.2 | 44.14±0.2 | 48.44±1.0 | 27.24±0.6 | 67.32±0.5 | 41.63±0.3 | 53.83±0.8 | 48.66±0.7 | 39.14±0.6 | 56.56±0.3 |
| AEVC | 40.57±2.7 | 24.67±2.6 | 45.25±0.3 | 22.10±0.2 | 69.51±3.0 | 41.27±0.7 | 61.54±1.4 | 64.13±1.1 | 51.39±1.0 | 69.67±0.3 |
| INMKC | 34.13±0.0 | 15.36±0.0 | 51.42±0.0 | 29.29±0.0 | 65.99±0.0 | 43.45±0.0 | 62.52±0.0 | 59.68±0.0 | 51.69±0.0 | 68.72±0.0 |
| ASCR | 24.11±1.2 | 10.59±1.3 | 40.23±0.8 | 26.88±0.2 | 63.12±0.4 | 51.40±0.8 | 43.34±3.2 | 51.86±2.4 | 45.76±1.3 | 69.53±0.6 |
| ALPC | 49.29±0.0 | 34.20±0.0 | 43.58±0.0 | 20.69±0.0 | 74.60±0.0 | 56.25±0.0 | 59.94±0.0 | 65.06±0.0 | 38.07±0.0 | 55.02±0.0 |
| **Ours** | **55.71±0.0** | **45.31±0.0** | **53.56±1.8** | **34.49±1.0** | **90.95±3.1** | **75.26±2.5** | **65.23±1.8** | **71.83±0.8** | **53.94±1.1** | 69.86±0.6 |

for MMC.

**Single-modal clustering methods**: K-Means (KM), Information Bottleneck (IB) (Tishby et al., 1999). These methods are applied to individual modalities separately, and we report the best performance among all single-modal results.

**Full-modal clustering methods**: All-modal K-Means (AmKM), All-modal IB (AmIB). These methods concatenate all modalities and directly apply single-modal clustering strategies.

**MMC methods**: LMSC (Zhang et al., 2017, CVPR), MLAN (Nie et al., 2018, TIP), GMC (Wang et al., 2020, TKDE), SMVSC (Sun et al., 2021, ACM MM), FPMVS-CAG (Wang et al., 2022, TIP), OMSC (Chen et al., 2022, KDD), FIMVC-VIA (Liu et al., 2024b, TNNLS), AEVC (Liu et al., 2024a, CVPR), INMKC (Feng et al., 2025, AAAI), ASCR (Xu et al., 2025, AAAI), and ALPC (Chen et al., 2025, AAAI).

### 4.3. Implementation Details

We implemented the proposed framework on a Windows 10 system equipped with an NVIDIA RTX 3080 GPU using MATLAB R2021a. All experiments were independently replicated 30 times, and we report the mean and standard deviation of all performance metrics. For fair comparison, single-modal clustering methods report their best performance across all modalities for each dataset. For MMC methods, we use the parameter settings reported in the original papers and present their best results under the same evaluation protocols.

For SGB-IB, $n_{min}$ is fixed at 20 and $\tau$ is fixed at 0.6. The

IB trade-off parameter $\beta$ is set to 500 as the default value. For $\alpha$, we perform a grid search over $[0, 1]$ with a step size of 0.1 to select the optimal value. Subsequent parameter and threshold sensitivity analyses confirm that our method is robust to these settings.

To ensure a comprehensive evaluation, Clustering Accuracy (ACC) and Normalized Mutual Information (NMI) are adopted as two widely used clustering metrics, where higher values indicate better clustering performance.

### 4.4. Clustering Results and Analysis

We conduct extensive experiments to benchmark our SGB-IB against the aforementioned clustering methods. Table 2 presents the clustering performance (mean ± standard deviation) of all methods on five datasets. SGB-IB achieves notable improvements over traditional clustering methods including KM, IB, AmKM and AmIB, and also outperforms other state-of-the-art multi-modal clustering approaches. This verifies the validity of combining GB learning and the IB framework.

Most methods converge steadily with minimal numerical variation across multiple independent runs. Several algorithms including our model exhibit zero standard deviation on certain datasets, indicating highly consistent results across repeated runs. This phenomenon is frequently observed among conventional clustering algorithms. Even when minor deviations emerge on other datasets, our method still maintains excellent stability. Collectively, these observations demonstrate that the promising performance of SGB-IB is consistently reproducible and robust to initialization across datasets.

*Table 3.* Ablation study on different multi-modal datasets.

| Methods | SOCCER | | Reuters | | BBC | | WVU | | COIL100 | |
|---|---|---|---|---|---|---|---|---|---|---|
| | ACC | NMI | ACC | NMI | ACC | NMI | ACC | NMI | ACC | NMI |
| (1) w/o GB&$S_{\mathcal{G}}$ | 53.64±2.8 | 43.73±2.9 | 50.71±4.8 | 31.01±3.2 | 81.29±10.3 | 72.89±7.3 | 62.03±5.1 | 63.77±3.3 | 52.17±0.0 | 69.19±0.0 |
| (2) w/o $S_{\mathcal{G}}$ | 54.97±0.2 | 44.92±0.1 | 52.84±0.8 | 33.41±1.7 | 87.64±7.6 | 73.04±4.8 | 62.44±5.3 | 68.11±2.0 | 52.86±1.0 | 69.42±0.8 |
| (3) Full Model | **55.71±0.0** | **45.31±0.0** | **53.56±1.8** | **34.49±1.0** | **90.95±3.1** | **75.26±2.5** | **65.23±1.8** | **71.83±0.8** | **53.94±1.1** | **69.86±0.6** |



| (a) | (b) | (c) | (d) | (e) |

*Figure 3.* Parameter analysis of SGB-IB on different datasets. (a)–(e) SOCCER, Reuters, BBC, WVU, COIL100.

On the SOCCER dataset, SGB-IB delivers favorable clustering results for images with subtle semantic distinctions. Recursive GB splitting generates multi-scale adaptive representations, capturing both fine visual details and high-level semantic structures. On the Reuters dataset, our method surpasses all comparison algorithms such as INMKC by exploiting internal linguistic consistency via structure-aware weights. Even on the large-scale COIL100 dataset, our method reaches an ACC of 53.94% and outperforms all baselines. This further demonstrates that SGB-IB can adapt well to large-scale multi-modal data while maintaining reliable clustering accuracy.

The experimental results summarize three core merits of SGB-IB. Specifically, adaptive structure-aware weights strengthen cross-modal complementary information. Recursive GB partitioning produces data-driven multi-granularity features. The IB constraint balances discriminative information retention and redundant feature compression.

### 4.5. Ablation Study

To quantify the individual and synergistic contributions of GB generation and $S_{\mathcal{G}}$ weighting mechanisms, we design three ablation variants: (1) w/o GB & $S_{\mathcal{G}}$, (2) w/o $S_{\mathcal{G}}$ (only GB generation retained), (3) Full Model. The results in table 3 validates the effectiveness of each core module of

SGB-IB.

Removing both GB and $S_{\mathcal{G}}$ yields the worst performance across all datasets, affirming the necessity of both components; retaining only GB generation brings improvements, with an ACC gain of 2.13% on Reuters, as this module exploits GB representations and captures locally coherent structures; integrating $S_{\mathcal{G}}$ delivers optimal performance, since it adaptively weights GB similarities to amplify inter-modal consensus. Together, these complementary mechanisms achieve synergy and significantly boost clustering performance.

### 4.6. Parameter Sensitivity

We analyze the parameters $\alpha$ and $\beta$ of our method, evaluating $\alpha$ within $[0, 1]$ at intervals of 0.1 and $\beta$ on the set $\{10, 50, 100, 500, 700, 900\}$. As shown in Figure 3, the proposed method maintains stable clustering accuracy across most datasets, with insensitivity to $\alpha \in [0, 0.4]$ and $\beta \in \{100, 500, 700, 900\}$ verifying its robustness to parameter variations. We thus select $\beta = 500$ as the default value, given its stable and favorable performance across all datasets.

### 4.7. Threshold Sensitivity

To validate the choice of fixed thresholds, we conduct a systematic sensitivity analysis on $n_{\min}$ and $\tau$, where the candidate values for $n_{\min}$ are 10, 15, 20, 25, 30 and those for $\tau$ are 0.3, 0.4, 0.5, 0.6, 0.7, 0.8, 0.9. Results are presented in Figure 4. The sensitivity analysis for $n_{\min}$ reveals that a setting of 20 yields stable ACC performance across most datasets, and the Reuters dataset attains its local ACC peak at this threshold. For $\tau$, setting it to 0.6 ensures that the ACC of all datasets remains relatively stable. Based on these observations, we fix $n_{\min} = 20$ and $\tau = 0.6$ as the

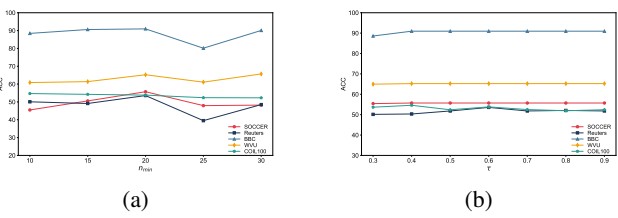

| (a) | (b) |

*Figure 4.* Threshold sensitivity. (a) $n_{\min}$; (b) $\tau$.

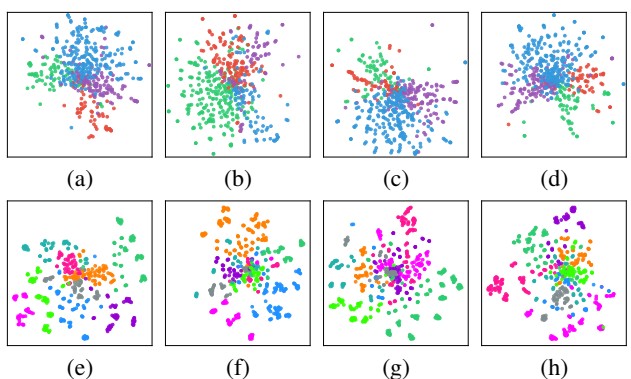

*Figure 5.* Visualization of clustering results for different modalities of the BBC and WVU datasets via t-SNE. (a)–(d) BBC Modal 1–4; (e)–(h) WVU Modal 1–4.

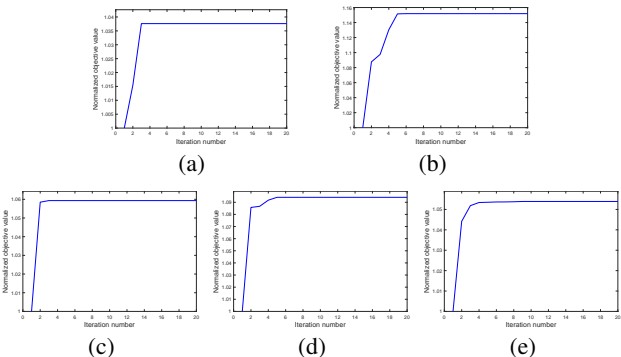

*Figure 6.* Convergence analysis of SGB-IB on different datasets. (a)–(e) SOCCER, Reuters, BBC, WVU, COIL100.

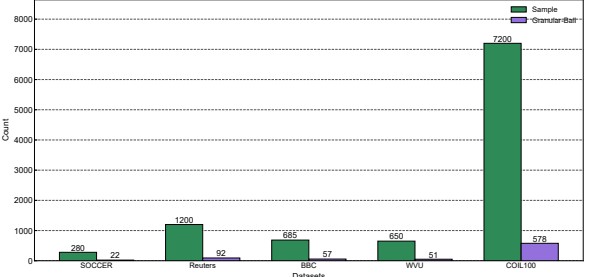

*Figure 7.* Count statistics of samples and granular-balls across datasets.

default settings for all experiments.

### 4.8. Visualization Validation

To further illustrate the effectiveness of our clustering results, we present t-SNE visualizations of the BBC and WVU datasets, as shown in Figure 5. The visualizations reveal that most modalities form compact intra-cluster structures with well-defined inter-cluster boundaries, with the first modality of both datasets exemplifying this characteristic most clearly. These observations demonstrate that our proposed GB generation mechanism and structure-aware optimization strategy effectively capture discriminative multi-modal features for each individual modality, thus yielding high-quality and visually coherent clustering performance across all modalities.

### 4.9. Convergence Analysis

Figure 6 presents the convergence behavior of SGB-IB across five benchmark datasets. To ensure consistent visualization, all objective values are normalized by their initial values, i.e., $\mathcal{L}_t/\mathcal{L}_0$. The curves show rapid convergence and stabilization within a small number of iterations, which can

be attributed to the effective initialization provided by GB partitioning. Such initialization provides a favorable starting point for the IB optimization. These results further verify the stability of the proposed SGB-IB algorithm.

### 4.10. Granular-Ball Count Statistics

To quantitatively characterize the adaptive granularity of SGB-IB, we report the final GB numbers and sample sizes for all datasets in Figure 7. The GB count varies from 22 to 578 and correlates well with dataset scales, demonstrating the data-adaptive granularity of our framework.

Specifically, the dual-threshold mechanism guarantees sufficient samples within each GB for statistical reliability and prevents over-segmentation. The structure-aware weight further adaptively regulates GB contribution, which ensures overall model stability even as GB count increases on complex datasets.

## 5. Conclusion

Our SGB-IB method achieves synergistic optimization of local structure preservation and global redundancy suppression for MMC through recursive GB splitting and structure-aware weighting mechanisms. This approach effectively balances fine-grained detail retention and coarse-grained structural integrity, demonstrates the effectiveness of the fused GB learning and information-theoretic framework.

SGB-IB is designed for general MMC tasks, yet it may encounter challenges with real-world data featuring missing modalities or misaligned modalities. The core coupling of GB structure awareness and the IB framework is task-agnostic, supporting natural generalization to diverse learning tasks. In future research, we will extend the proposed framework to handle missing modalities and misaligned modalities, adapt the method for dynamic MMC scenarios, and conduct comprehensive validation of its generalization across multi-task learning, transfer learning, and continual learning. These efforts will further improve its applicability to large-scale, complex, dynamic multi-modal datasets.

## Acknowledgement

This work was supported by National Natural Science Foundation of China under Grant 62576320, Henan Province Outstanding Youth Science Fund Program under Grant 252300421223.

## Impact Statement

This paper presents work whose goal is to advance the field of Machine Learning. There are many potential societal consequences of our work, none which we feel must be specifically highlighted here.

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
