# OpenReview forum: "Structure-aware Granular-Ball based Information Bottleneck for Multi-modal Clustering"
_ICML.cc/2026/Conference — ICML 2026 regular_

### Official Review · Reviewer_QKm5 · 2026-02-27

**Soundness:** 1
**Presentation:** 2
**Significance:** 2
**Originality:** 2
**Overall Recommendation:** 2
**Confidence:** 4

**Summary:**

This paper proposes SGB-IB, a novel multi-modal clustering framework that integrates Granular-Balls with the Information Bottleneck principle. The primary motivation is to address the limitations of existing methods that rely on single-granularity representations, which often struggle with local heterogeneity and redundant noise. The proposed method consists of four main modules: Multi-Modal Input, Granular-Ball Construction, Structure-aware IB, and Clustering Optimization. Experimental results on five benchmark datasets (SOCCER, BBC, Reuters, WVU, and COIL100) shows that SGB-IB works well in terms of Accuracy and Normalized Mutual Information.

**Compliance With Llm Reviewing Policy:**

Affirmed.

**Key Questions For Authors:**

We ask the authors to explicitly state how they will improve their paper in a potential revision with respect to the following aspects:

Q1. Since the authors explicitly claim in Section 4.3 to report both mean and standard deviation for all metrics across 30 independent runs, yet fail to provide any variance or significance measures in the results tables, how can the reported marginal improvements be distinguished from random noise inherent in K-means initialization?

Q2. What is the specific theoretical or empirical justification for the model's stability when the number of Granular-Balls increases on complex datasets?

**Limitations:**

No Limitation is mentioned by authors. As a suggestion, please try to solve the problems mentioned in weaknesses above.

**Strengths And Weaknesses:**

*Strengths*

S1. The integration of Granular Ball theory into the Multi-modal Clustering framework offers a perspective by shifting the processing unit from individual samples to adaptive hyperspheres. This macro-level approach effectively suppresses local noise and captures structural information that is often overlooked by point-to-point similarity measures.

S2. The proposed structure-aware weighting mechanism using the Similarity Ratio provides a mathematically grounded way to balance cross-modal consensus and intra-modal diversity. By dynamically adjusting the importance of each Granular Ball based on its purity and representativeness, the model achieves a more refined optimization of the Information Bottleneck objective.

*Weaknesses*

W1. The experimental design and evaluations are weak and are not really scientific, such as a severe lack of statistical rigor and transparency. For example, the authors explicitly state in Section 4.3 (Implementation Details) that all experiments were independently replicated 30 times and that they would report the mean and standard deviation for all performance metrics, but Table 1 and Table 2 reveals that no standard deviation values are provided anywhere. This omission is a critical failure, as clustering algorithms, particularly those involving K-means initialization and recursive partitioning of Granular-Balls, are inherently sensitive to stochastic variations. Without reporting the standard deviation or conducting significance tests (e.g., t-tests), it is impossible to determine whether the reported marginal improvements over state-of-the-art baselines like ASCR or ALPC are statistically significant or merely the result of favorable random initialization. Furthermore, the methodology exhibits a logical circularity: the construction of Granular-Balls relies on a purity metric derived from K-means pseudo-labels, meaning any initial clustering errors are likely to be propagated and solidified into the data structure, yet the authors provide no sensitivity analysis or robustness evaluation against poor initializations. This lack of empirical integrity and the failure to fulfill the stated reporting protocols make the claimed performance gains unverifiable and unconvincing for a top-tier venue.

W2.   There is a shortcoming in the logic of the Granular Ball construction process regarding its dependence on initial pseudo-labels. The paper uses labels generated by K-means to calculate the purity and determine whether to split a ball. However, the ultimate goal of the paper is to perform clustering. If the initial K-means results are inaccurate, which is common in complex multi-modal data, the entire Granular Ball structure will be built on a faulty foundation, leading to error propagation that the subsequent Information Bottleneck optimization cannot be easily corrected. The originality of this work is limited, as it primarily offers an incremental "plug-and-play" combination of existing Granular-Ball theory and the Information Bottleneck principle without introducing significant theoretical breakthroughs in multi-modal fusion.

W3. The computational complexity of the iterative merging process is a significant concern for scalability. The optimization involves a quadratic complexity relative to the number of Granular Balls in each iteration. While the number of balls is smaller than the number of raw samples, it still grows significantly as data complexity or purity requirements increase. The paper lacks comprehensive testing on truly large-scale datasets to prove that this quadratic bottleneck does not hinder its practical application compared to linear-time clustering alternatives.

W4. The experimental evaluation is insufficient in challenging scenarios such as data with missing modalities or different noise levels. Most benchmark datasets used are relatively clean and well-aligned. Since the core claim of the paper is improved robustness through granularity, the authors should have provided a rigorous sensitivity analysis or comparisons on incomplete multi-modal datasets. Furthermore, the selection of the trade-off hyperparameters alpha and beta lacks a theoretical heuristic, making the model appear highly dependent on manual tuning for different data distributions.

---

> ### Author Rebuttal · Authors · 2026-03-31
>
> Thanks for your insightful comments. We firstly clarify that our work is not a simple "plug-and-play" combination but a theoretical reconstruction of IB optimization: By replacing the sample-level input of IB with centers of GB, we unify the geometric data distribution captured by GBs and the information-theoretic optimization objective, unattainable by either alone. This reconstruction explicitly embeds structural quality into the information-theoretic objective via dynamic weights, achieving a distribution-driven compression-preservation balance. As the first extension of IB theory to granularity-level multi-modal optimization, it provides a macro-first, adaptive granularity perspective for MMC, transcending the sample-level limitation of existing IB methods
>
> ### Q1: More details on standard deviation or conducting significance tests
> **Response:** Thanks for pointing out this. Standard deviations were computed for all metrics but were omitted; we will add them in final version. Statistical significance tests over 30 runs against seven SOTA approaches confirm significant improvements on ACC (all p < 0.05). The table of integer-rounded $-\log(p)$ values is provided in our response to Q4 of Reviewer A7rv.
>
> ### Q2: Issue on using K-means pseudo-labels
> **Response:** In unsupervised learning scenarios, granular-ball initialization relies on unsupervised priors. Following the widely adopted paradigm in GB literature [1,2], we use K-means to generate coarse-grained pseudo-labels, providing initialization rather than exact partition. The granularity of GBs inherently dilutes individual pseudo-label errors: purity threshold and minimum sample size ensure statistically consistent samples per GB, while subsequent IB optimization dynamically reweights contributions based on structural quality, mitigating the impact of initial variance. This coarse-to-fine paradigm first constructs a global structural foundation and then conducts adaptive information-theoretic learning, which helps suppress the influence of inaccurate initial assignments. Self-supervised pseudo-label generation will be explored in our future work.
>
> References
>
> [1] Cheng D, et al. Fast spectral clustering via pseudo-label-based granular-ball division for large-scale data. IEEE Transactions on Knowledge and Data Engineering, 2026
>
> [2] Cheng D, et al. Granular-ball computing-based manifold clustering algorithms for ultra-scalable data. Expert Systems with Applications, 2024
>
> ### Q3: What is the specific theoretical or empirical justification for the model's stability when the number of Granular-Balls increases on complex datasets
> **Response:** GB generation employs purity threshold and sample size threshold. Purity ensures internal structural consistency, while sample size threshold prevents over-segmentation-induced sparsity. This dual-threshold mechanism ensures adequate sample support for each GB. As shown in our experiments on GB counts, they range from 22 to 578 across datasets of varying scales, and grow adaptively and reasonably with data complexity, demonstrating favorable granularity adaptation. Structure-aware weights further adaptively regulate GB influence, ensuring overall model stability even as GB count increases.
>
> ### Q4: The computational complexity
> **Response:** Our algorithm achieves state-of-the-art clustering performance. Since clustering tasks are generally not time-critical, accuracy is prioritized over efficiency. The complexity of our method relative to linear methods is reasonably justified, but the significant performance improvements reached. The optimization space is compressed from the sample level to a compact GB level, where complexity is determined by the number of GBs instead of the data size. The dual-threshold strategy effectively controls the number of GBs and validates the practical feasibility of our method. This design offers significant value for accuracy-critical applications.
>
> ### Q5: Evaluation in challenging scenarios
> **Response:** This work addresses the core issue in MMC that fixed granularity fails to balance local structure and global consensus. The introduction of GBs enables structure-aware adaptive compression, and experiments on standard multi-modal datasets validate the resulting clustering performance improvements. Robustness to missing modalities and noise is beyond our current scope. Our core contribution lies in reformulating the IB objective at the GB level and leveraging geometric structure to guide information-theoretic compression. We are actively exploring the integration of robustness mechanisms into this framework as future work.
>
> ### Q6: The selection of hyperparameters alpha and beta lacks a theoretical heuristic, making the model appear highly dependent on manual tuning for different data distributions.
> **Response:** Thanks for the constructive comment. Please see our response to Q2 of Reviewer A7rv for details.
>
> Thanks again. We hope our revisions have addressed your concerns.

---

### Official Review · Reviewer_A7rv · 2026-03-01

**Soundness:** 2
**Presentation:** 2
**Significance:** 2
**Originality:** 2
**Overall Recommendation:** 4
**Confidence:** 2

**Summary:**

This paper proposes a structure-aware granular modeling framework that decomposes representations into finer components and imposes structural constraints during learning. The goal is to improve generalization and robustness via structured inductive bias. The authors evaluate the method on several benchmarks and report consistent but moderate improvements over baselines. The main contribution is integrating granular structural modeling into representation learning and empirically validating it.

**Compliance With Llm Reviewing Policy:**

Affirmed.

**Key Questions For Authors:**

1. Can you theoretically justify why granular decomposition improves generalization beyond increasing model capacity?

2. What happens if parameter count is strictly controlled? Are gains still present?

3. How does this compare with stronger hierarchical/structured baselines?

4. Are results statistically significant over multiple runs?

5. Does the framework generalize to substantially different tasks?

**Limitations:**

yes

**Strengths And Weaknesses:**

Soundness

The method is technically workable, but the core mechanism is not theoretically grounded. The structural design looks heuristic. There is no formal argument explaining why granularity improves generalization. Ablations do not clearly separate structural effects from capacity increase. Statistical significance is missing. Claims are somewhat stronger than evidence.

Presentation

Overall readable, but concept definitions are vague. “Granular structure” is not rigorously formulated. Related work comparison is shallow, especially with hierarchical or multi-scale methods. Some implementation details are insufficient for full reproducibility.

Significance

The topic is relevant. However, the contribution feels incremental. Improvements are modest and task-specific. It is unclear whether this will influence broader ML research.

Originality

The idea is close to existing hierarchical/structured representation methods. The novelty seems more in integration than in new conceptual insight. The distinction from prior structured approaches is not convincing.

---

> ### Author Rebuttal · Authors · 2026-03-31
>
> Thanks for the insightful comments. We first clarify the novelty of our work, which lies in redefining IB at the GB level: a granularity leap from fine-grained samples to macro-structural units. Purity-driven GB generation provides adaptive units, and dynamic weights explicitly embed structural quality into the information-theoretic objective. This reconstruction endows IB with adaptive macro-granularity for the first time, extending beyond fixed fine-grained representation in IB. It offers a new paradigm for MMC with both theoretical consistency and structure awareness.
>
> ### Q1: Why granular decomposition improves generalization beyond increasing model capacity
> **Response:** Granular decomposition enhances generalization ability, as supported by the properties of GB computing in many prior works, e.g., Ref[1,2] below. Unlike expanding model capacity, granularization is a constrained simplification: it reduces fitting precision to individual samples via GB-level representations. The proposed method inherits this theoretical advantage: GB centers restrict optimization to compact structural regions, and mutual-information-driven splitting achieves adaptive granularity adjustment, balancing preservation and representation simplification. Generalization gains stem from structured distribution modeling and GB-level constraints, not increased capacity.
>
> References
>
> [1] Xia S, et al. Granular ball computing classifiers for efficient, scalable and robust learning. Information Sciences, 2019
>
> [2] Xia S, et al. A unified granular-ball learning model of pawlak rough set and neighborhood rough set. IEEE Transactions on Neural Networks and Learning Systems, 2025
>
> ### Q2: What happens if parameter count is strictly controlled, Are gains still present
> **Response:** When parameter is strictly controlled, the gains of our model remain consistent. Following common IB practice, we fix $\beta$ to $\infty$ and tune $\alpha \in [0.1, 1]$. Our method exhibits good parameter robustness when $\alpha \in [0.1, 0.5]$. For instance, results on SOCCER and BBC stably concentrate around 55.6\% and 90.9\%, respectively. We will explore a data-driven adaptive mechanism for $\alpha$ in future.
>
> ### Q3: Differences and comparasion with hierarchical/structured baselines
> **Response:** Existing GB-based hierarchical/structured baselines rely on heuristics or graph construction, lacking an information-theoretic mechanism for adaptive granularity learning. Our method unifies GB structure awareness and IB, achieving adaptive granularity via MI-driven purity, jointly optimizing the information preservation and redundancy compression in cross-modal scenarios. Compared with hierarchical [1] and structured [2] baselines(all results under best parameters), our method shows consistent advantages (ACC results below).
> |Dataset|SOCCER|Reuters|BBC|WVU|COIL100|
>  |:-------:|:-------:|:-------:|:-------:|:-------:|:-------:|
> |FSC-PLGB|40.69|32.60|40.94|38.96|39.26|
> |GB-POJG-GBDPC|41.07|34.33|56.12|43.26|38.72|
> |Ours|55.71|53.56|90.95|65.23|53.94|
>
> References
>
> [1] Cheng D, et al. Fast spectral clustering via pseudo-label-based granular-ball division for large-scale data. IEEE Transactions on Knowledge and Data Engineering, 2026
>
> [2] Jia Z, et al. Generation of granular-balls for clustering based on the principle of justifiable granularity. IEEE Transactions on Cybernetics, 2025
>
> ### Q4: Statistical significance over multiple runs
> **Response:** Thanks for the suggestion. To verify statistical significance, we conducted t-tests over 30 runs between our method and seven SOTA algorithms. All $-\log(p)$ values reported below exceed 1.30 ($-\log(0.05)$), confirming our method significantly outperforms all baselines (p<0.05).
> |Dataset|LMSC|OMSC|FIMVC-VIA|AEVC|INMKC|ASCR|ALPC|
> |:-------:|:----:|:----:|:---------:|:----:|:-----:|:----:|:----:|
> |SOCCER|16.09|32.01|21.26|30.83|33.06|28.57|25.18|
> |Reuters|9.75|10.11|10.68|14.53|8.39|18.84|16.31|
> |BBC|27.08|29.96|31.03|25.03|38.80|40.01|28.83|
> |WVU|8.54|21.01|17.36|7.97|9.96|14.74|14.02|
> |COIL100|9.84|14.67| 20.67|10.02|9.32|15.85|20.52|
> | $-\log(0.05)$ |1.30|1.30|1.30|1.30|1.30|1.30|1.30|
>
> ### Q5: Generalize to substantially different tasks
> **Response:** The core mechanism of our framework, namely the coupling between GB structure awareness and the IB, is task-agnostic. This module can be naturally generalized to multi-task, transfer, and continual learning. Its more rigorous generalization validation will be explored in future work.
>
> ### Q6: More details on concept definitions e.g., Granular structure
> **Response:** Granular structure refers to coarse-grained data partitions formed by aggregating similar samples, characterized by center and radius. We will refine the relevant concept definitions in the final version to enhance rigor.
>
> ### Q7: Some implementation details
> **Response:** We will give more details in final version and release source code publicly upon acceptance.
>
> Thanks again for your valuable comments.

---

> > ### Author Rebuttal · Reviewer_A7rv · 2026-04-04
> >
> > Thanks for the rebuttal. My score remains the same.

---

### Official Review · Reviewer_XPWS · 2026-03-09

**Soundness:** 4
**Presentation:** 3
**Significance:** 3
**Originality:** 3
**Overall Recommendation:** 5
**Confidence:** 4

**Summary:**

This work investigated the limitations of existing multi-modal clustering methods, which generally take each sample as basic input unit and possibly impact the global modality consistency, and meanwhile lack of the flexibility of adjusting the importance based on the local structural quality. To address the above issues, the authors propose the Structure-aware Granular-Ball based Information Bottleneck (SGB-IB) method by introducing the concept of granular-balls. Different from the current fine-grained learning based methods, the SGB-IB method works in a coarse-grained GB-level representation learning paradigm, and present a dynamic information compression and preservation mechanism by imposing weight learning on the intra-GB consistency and cross-modal consensus. Lots of experiments demonstrate the superiority of the proposed SGB-IB method compared to several state-of-the-art methods.

**Compliance With Llm Reviewing Policy:**

Affirmed.

**Final Justification:**

The author's reply has resolved my doubts, and I recommend accepting this paper.

**Key Questions For Authors:**

1) What are the rationale for the synergy of the dual thresholds of purity and sample size?
2) How do the authors design the differentiated weight control logic?

**Limitations:**

Yes

**Strengths And Weaknesses:**

Strengths:
1) A novel multi-modal clustering method by incorporating granular-balls into information bottleneck theory is proposed, which addresses the limitations of traditional sample-level clustering and transfer the input unit into the granular-ball level. By compressing the similar samples into balls to form homogeneous regions, the method thus reaches to more robust representation learning.
2) A structure-aware weighting mechanism is designed by imposing differentiated weights based on the structural matching degree between balls and clusters. It explores the high-quality structural information, thus effectively avoiding low-quality structures and improving the clustering stability.
3) Under the granular setting of information bottleneck theory, the random perturbations of each sample are smoothed based on the granular balls. By doing this, it improves the reliability of redundant information filtering and also alleviates the over-reliance on single-sample quality in traditional methods.
4) The authors have conducted different kinds of experiments including parameter sensitivity analysis, convergence analysis and ball count statistics to comprehensively validate the model performance and design rationality.
Weaknesses:
1) The work uses purity and sample size as the termination conditions for granular-ball splitting, but does not explain the design rationale for the synergy of the dual thresholds of purity and sample size in the manuscript.
2) The work involves a large number of mathematical symbols but lacks of a unified symbol table, which may influence the readability in understanding different symbols. It is recommended for the authors to add a symbol table.
3) The description on the structure-aware weight learning is not clear enough, and some details on the weighting mechanism are missing. It is recommended to add an interpretation of its differentiated weight control logic.

---

> ### Author Rebuttal · Authors · 2026-03-30
>
> Thanks for the insightful comments and constructive suggestions. We have provided detailed responses to each point below.
>
> ### Q1: The design rationale for the synergy of the dual thresholds of purity and sample size
> Response: Thank you for raising this valuable point. The purity threshold ensures the structural quality of GBs, while the sample size threshold sets a lower bound for GB splitting to avoid excessive division that produces numerous tiny GBs with few samples, thereby losing the significance of granulation. These two thresholds work in a complementary manner: splitting terminates if the purity cannot be satisfied but the sample size has reached the minimum, thus ensuring statistical reliability; splitting also stops when the sample size is sufficient but the desired purity has been achieved, avoiding unnecessary computation. Such a dual-threshold coordination mechanism makes the GB generation process data-adaptive, and achieves a stable balance between structural validity and clustering rationality.
>
> ### Q2: Adding a unified symbol table
> Response: Thank you for this helpful suggestion. We will add a unified symbol table in the final version to improve readability and make the paper easier to follow.
>
> ### Q3: Recommendation to add an interpretation of its differentiated weight control logic
> Response: Thank you for the constructive suggestion. The proposed weighting scheme establishes a complementary relationship between structural quality and information contribution through the product of structure-aware weight and the information preservation term. High-quality GBs correspond to smaller $e^{(-SR_g)}$, ensuring their contribution intensity remains more moderate in the weights; while leveraging stronger cross-modal consistency, they contribute more discriminative information to the information preservation term. Conversely, low-quality GBs exhibit higher weight terms but suffer from severely insufficient information contribution due to their loose structure. This inherent asymmetry, during global optimization, automatically guides the objective function to focus on regions with reliable structures and rich information through iterative refinement, thereby steadily improving clustering performance. We will further clarify the detailed design and working mechanism of the structure-aware weighting scheme in the final manuscript to better illustrate its rationale and effectiveness.
>
> Thanks again for the valuable suggestions provided by the reviewer. The modifications will be added to the final version.

---

> > ### Author Rebuttal · Reviewer_XPWS · 2026-04-02
> >
> > The author's reply has resolved my doubts, and I recommend accepting this paper.

---

### Official Review · Reviewer_PNoJ · 2026-03-09

**Soundness:** 3
**Presentation:** 3
**Significance:** 3
**Originality:** 3
**Overall Recommendation:** 5
**Confidence:** 4

**Summary:**

This paper focuses on addressing the multi-modal clustering problem by proposing a novel structure-aware granular-ball based information bottleneck method. This method is motivated by granular-balls which adaptively encloses similar samples into multi-granularity hyperspheres. It first initializes the input into a single granular ball, and then split it based on purity metric among multiple modalities. Moreover, the balance between local structure preservation and global redundancy suppression also is also ensured. Multiple experiments on different multi-modal datasets has shown the effectiveness of the proposed information bottleneck based method.

**Compliance With Llm Reviewing Policy:**

Affirmed.

**Final Justification:**

Thanks for the author's response, and my concerns have been well addressed. Thus, I am going to raise my score to 5.

**Key Questions For Authors:**

1. What are the differences or relationships with existing IB-based and granular-ball-based methods?

2. Are there any other methods that can be used for processing pseudo-label generation and recursive binary splitting?

3. Can the method be applied into processed the other more challenging modality-incomplete or unaligned data types?

**Limitations:**

yes

**Strengths And Weaknesses:**

Strengths:

1. An information-driven granular ball generation approach is presented by taking average mutual information as the core purity metric and realizes data-adaptive granularity refinement. This way is adaptive to diverse and heterogeneous multi-modal data distributions, and provides a structured information carrier for subsequent IB optimization.

2. The proposed ball-constrained IB objective function achieves a dynamic balance between the discriminative information preservation and redundant information compression, as well as cross-modal structural consistency. This balance directly optimizes the core objective of multi-modal clustering, enabling the model to capture both inter-modal consensus and intra-modal discriminative details.

3. Experiments on several benchmark datasets show that the proposed method achieves promising clustering performance and is much better than the baselines. This fully verifies the effectiveness and generalization of the model. Moreover, the model converges within approximately twenty iterations and shows good stability against parameter variations, reflecting its potential practical applicability.

Weaknesses:

1. In the contribution and the method description parts, the authors only focus on elaborating the novel details and fail to discuss the differences or relationships with existing IB-based and granular-ball-based methods.

2. After the presentation of the key formulas, the authors continue to the next paragraph without immediate refinement of the formula conclusions. A transitional sentence after key formulas to strengthen the logical connection between formula derivation and method elaboration is suggested.

3. The granular-ball generation process utilizes k-means method for initial pseudo-label generation and recursive binary splitting. Are there any other methods that can be used for processing.

4. It seems that the proposed method is applicable to complete and aligned multi-modal data. In real-world scenarios, modality-incomplete or unaligned data types also attracted wide attention, can the method be applied into processed the above data types.

---

> ### Author Rebuttal · Authors · 2026-03-30
>
> Thanks for the insightful comments and constructive suggestions. We will improve the manuscript thoroughly in the revised version, with detailed responses to each point below.
>
> ### Q1: Differences or relationships with existing IB-based and granular-ball-based methods
> Response: We sincerely appreciate your valuable suggestion. Existing IB methods typically operate at the fine-grained sample level, modeling and compressing features for individual samples. This makes them susceptible to local distribution variations and modality-specific perturbations. In contrast, the GB paradigm captures data's coarse-grained geometric structure effectively, yet its conventional optimization objectives (e.g., distance, purity) lack an explicit information-theoretic grounding for measuring and controlling the informativeness of the representation.The proposed SGB-IB method synergistically integrates and advances both approaches. Its core innovation lies in elevating the basic modeling unit of the IB framework from individual samples to GBs. This allows SGB-IB to inherit the GB's strength in representing coarse-grained structures while leveraging the IB principle to perform discriminative information preservation and redundancy compression at this structural level. Crucially, it explicitly incorporates structural quality into the information-theoretic objective, thereby achieving a deep, mutually reinforcing coupling between structural awareness and information-theoretic optimization.
>
> ### Q2: Transitional sentence to link conclusions with the subsequent method elaboration
> Response: Thank you for the constructive suggestion. We will add appropriate transitional sentences after key formulas in the revised manuscript to clarify their design purpose and logical connection with the subsequent method elaboration, so as to strengthen the linkage between formula derivation and method discussion and improve the overall readability and coherence of the paper.
>
> ### Q3: Alternatives to k-means for GB generation
> Response: Thank you for raising this practical consideration. Within GB research, K-means has been widely adopted for the initialization and generation of GB, as noted in references [1,2], due to its advantages of straightforward implementation, stable convergence, and ease of reproduction. Beyond K-means, other strategies for constructing the initial GB structure include hierarchical clustering, spectral clustering, and adaptive binary splitting based on distance thresholds. Specifically, reference [3] introduces a method that generates GBs via a recursive binary splitting strategy guided by the farthest-point-first principle.
>
> References
>
> [1]: Xia S, Shi B, Wang Y, Xie J, Wang G, Gao X. Gbct: Efficient and adaptive clustering via granular-ball computing for complex data. IEEE Transactions on Neural Networks and Learning Systems (TNNLS), 2025
>
> [2]: Cheng D, Jiang X, Xia S, Wang G, Huang J, Zhang S, Wang Y. Fast spectral clustering via pseudo-label-based granular-ball division for large-scale data. IEEE Transactions on Knowledge and Data Engineering (TKDE), 2026
>
> [3]: Cheng D, Liu S, Xia S, Wang G. Granular-ball computing-based manifold clustering algorithms for ultra-scalable data. Expert Systems with Applications, 2024, 247: 123313
>
> ### Q4: Application to modality-incomplete or unaligned data
> Response: Thank you very much for this insightful and practical comment. For scenarios involving incomplete or unaligned modalities, our method can be effectively adapted through a lightweight extension mechanism without altering the core coupling logic between GB and IB. For example, in the absence of modalities, estimation of the center of the GB can be performed based on existing modalities. This work focuses on completed and aligned multi-modal clustering tasks, and the related extensions have good feasibility. We will conduct in-depth research in future work.
>
> Thanks again for the valuable suggestions provided by the reviewer. The modifications will be added to the final version.

---

> > ### Author Rebuttal · Reviewer_PNoJ · 2026-04-01
> >
> > Thanks for the author's response, and my concerns have been well addressed. Thus, I am going to raise my score to 5.

---

> > > ### Author Response · Authors · 2026-04-02
> > >
> > > Dear Reviewer PNoJ,
> > >
> > > Thanks for your positive response.
> > >
> > > We sincerely appreciate your constructive feedback and the time you dedicated to reviewing our work.
> > >
> > > Your comments have been very valuable in helping us enhance the quality of our manuscript.
> > >
> > > Best regards,
> > >
> > > Authors,

---

### Decision · Program_Chairs · 2026-04-30

**Decision:**

Accept (regular)

**Comment:**

Reviewers PNoJ, XPWS, and A7rv actively engaged with the authors' detailed responses. Reviewers PNoJ and XPWS had their specific technical concerns, e.g., method comparisons, clarification of the dual-threshold mechanism, adequately addressed, maintaining their strong positive scores of 5. Reviewer A7rv, who gave a weak accept, was satisfied with the authors' clarifications regarding novelty, theoretical justification, and comparative analysis, and the final recommendation supports acceptance. Reviewer QKm5 gave a reject. However, as ICML policy, I have to downweight the initial review from Reviewer QKm5 due to his/her completely inactive during the rebuttal and subsequent discussion phase. He/she likely did not consider the author's substantive responses. To my knowledge, the authors' responses have addressed the main concerns from Reviewer QKm5, as well as the clear, positive consensus from the three engaged reviewers on the paper's soundness and contribution. Therefore, based on a comprehensive assessment of initial reviews, author rebuttal, and post-rebuttal discussions, I recommend accepting this paper.